# Electrosprayed Agar Nanocapsules as Edible Carriers of Bioactive Compounds

**DOI:** 10.3390/foods11142093

**Published:** 2022-07-14

**Authors:** Barbara Tomadoni, María José Fabra, Daniel Alexander Méndez, Antonio Martínez-Abad, Amparo López-Rubio

**Affiliations:** 1Grupo de Materiales Compuestos Termoplásticos (CoMP), Instituto de Investigaciones en Ciencia y Tecnología de Materiales (INTEMA), Universidad Nacional de Mar del Plata (UNMdP) y Consejo Nacional de Investigaciones Científicas y Técnicas (CONICET), Av. Colón 10850, Mar del Plata 7600, Argentina; btomadoni@gmail.com; 2Packaging Group, Food Safety and Preservation Department, Institute of Agrochemistry and Food Technology (IATA-CSIC), Catedrático Agustín Escardino Benlloch 7, 46980 Paterna, Spain; mjfabra@iata.csic.es (M.J.F.); daamendezre@iata.csic.es (D.A.M.); conaba@iata.csic.es (A.M.-A.); 3Interdisciplinary Platform for Sustainable Plastics towards a Circular Economy-Spanish National Research Council (SusPlast-CSIC), 26006 Madrid, Spain; 4Grupo de Investigación Bioecono, Facultad de Ciencias Económicas y Administrativas, Universidad del Tolima, Tolima 730006, Colombia

**Keywords:** agar, nanocapsules, electrospraying, chlorophyllin

## Abstract

Electrosprayed agar nanocapsules were developed using an acetic acid solution as solvent. The role of solution properties (viscosity, surface tension, and conductivity) in the formation of agar particles was assessed, together with the effect of both agar and acetic acid concentrations on the size and morphology of the resulting particles. Agar solutions with a concentration below 10% *w*/*v* were not suitable for electrospraying. Furthermore, the agar–acetic acid ratio was also critical for the formation of agar nanostructures (with an optimum ratio of 1:2). A decrease in particle size was also observed when decreasing agar concentration, with particle diameter values ranging between 50 and 400 nm. Moreover, the suitability of the electrosprayed agar nanocapsules as carriers for a model bioactive compound, chlorophyllin sodium copper salt (CHL), was also evaluated. The release profile of encapsulated CHL, with an estimated encapsulation efficiency of around 40%, was carried out in food simulants with different hydrophilicity (10% *v*/*v* and 50% *v*/*v* ethanol). While the release of the bioactive was negligible in the hydrophilic food simulant, an initial burst release followed by a slower sustained release was observed when the capsules were immersed in 50% ethanol solution. The results open up a broad range of possibilities that deserve further exploration related to the use of these edible polysaccharide-based nanocapsules.

## 1. Introduction

The incorporation of bioactive compounds to food products with the aim of extending their shelf-life or improving their nutritional quality is of high interest for the food industry [1]. Antioxidants, antimicrobials, probiotics, pigments, and flavoring agents are typically present in food products in small amounts, but are often highly active [2]. These active compounds are usually difficult to process and store, since they can impart strange aromas, are prone to rapid degradation, and can interact with other compounds in the food matrix, leading to a loss in the quality of functional assets. For example, phenolic compounds that are used as natural antioxidants for their health benefits can decrease their bioavailability after binding with carbohydrates and proteins present in the food matrix [3]. Natural pigments are also affected by other components naturally occurring in foods, especially after processing steps that damage the structure of the product, releasing compounds such as acids and enzymes. These compounds can interact with the natural pigments with a subsequent unwanted change in the color [4]. In this sense, encapsulation is a powerful tool to overcome many of the previously mentioned drawbacks, because it allows the protection of a great variety of compounds accomplished by covering the active molecules with a protective matrix. Encapsulation can also reduce off-flavors from certain vitamins and minerals, allow sustained release of nutrients, improve stability under high-temperature and -humidity conditions, and reduce undesirable chemical interactions with other ingredients, among others.

An interesting technique to develop micro- or nanostructures for delivery of active principles is electrohydrodynamic processing (EHD), which includes two variations of the technique: electrospinning and electrospraying. These are processing technologies where a polymer solution can be either spun or sprayed through the application of an electric field, giving raise to fibers or particles, respectively [5]. These are economical and industrially scalable technologies with potential applications in the food area [6,7]. Their main uses in the food industry include encapsulation of bioactive compounds, immobilization of enzymes, materials for packaging, food coatings, and materials for filtration processes [7]. Another important advantage of EHD techniques is that they offer the possibility of producing high-performance nano- or microfibers and nano- or microcapsules, developing materials with a high surface–volume ratio. Electrohydrodynamic process parameters, i.e., applied potential, electric field, spin distance, and flow velocity, as well as fluid properties, i.e., pH, conductivity, viscosity, surface tension, and dielectric constant, can be adjusted to optimize the morphology and the characteristics of the structures obtained.

Capsules or fibers produced through electrospinning or electrospraying also have an interesting advantage regarding their functionality, since these structures are suitable matrices for slow release of bioactive compounds [7]. EHD techniques would enable encapsulating active principles that are thermosensitive and, hence, would not be suitable for other encapsulation methods involving high temperatures, such as spray-drying. Overall, the structures obtained by means of electrospraying can provide a good vehicle for the encapsulation or immobilization of different bioactive compounds without the loss of their activity or specificity [8]. Moreover, electrospraying and electrospinning are ideal for development of materials for food applications, since these processing techniques allow the use of food-grade biopolymers [7]. Food-grade biopolymers such as polysaccharides [9,10] or proteins [9,11] have been used to produce nano- and microstructures by electrospinning or electrospraying [12]. For example, alginate and zein were used as polymers for the microencapsulation of probiotic *Lactobacillus acidophilus* through electrospraying [13], while gelatin-coated κ-carrageenan capsules for delivery of phenolic compounds were also produced through EHD [14].

Amongst the biodegradable materials from renewable sources which can be used for encapsulation purposes, agar, a phycocolloid extracted from some red seaweed species, is one of the most interesting ones. This is mainly due to its thermoplastic, biodegradable, and biocompatible characteristics, and because it shows great mechanical strength with moderate water resistance and good barrier properties, very relevant in terms of protection [15]. Agar has been previously studied as a polymer matrix for slow release of active compounds [16,17]. However, there are few studies of agar-containing structures obtained through electrohydrodynamic techniques, and, to the best of our knowledge, there is no information related to the production of pure agar micro- or nanocapsules by means of the electrospraying technique. In a previous study, Sousa et al. [18] successfully obtained agar-based nanofibers by means of the electrospinning technique through blending it with polyvinyl alcohol (PVA). These authors concluded that the addition of PVA was crucial to improve the rheological properties of the solution, thus allowing to process it electrohydrodynamically. These authors also studied the electrospinnability of agar/PVA blends, but with deep eutectic solvent (DES), which allowed them to increase the content of the phycocolloid [19].

Studies of nano- and microstructures of phycocolloids obtained by electrohydrodynamic processes are scarce, especially based on agar. The poor electrosprayability or electrospinnability of pure agar solutions in water could be related to the fact that the concentration that provides a suitable rheology for EHD processing is very low and, hence, does not provide sufficient chain entanglements [7]. To overcome this issue, this study focuses on improving the electrosprayability of agar with organic solvents such as acetic acid. This acid previously showed hydrolysis of agar [20]. This could provide suitable chain sizes, small enough to allow a suitable viscosity for EHD processing at higher polymer concentrations, but long enough to allow chain entanglements and the formation of the capsules [7]. The use of agar as matrix to produce nanofibers or nanocapsules could open up new and interesting applications of this biopolymer, not yet explored [18].

In this particular study, chlorophyllin sodium copper salt (CHL) was selected as a model bioactive compound to be incorporated into agar nanocapsules. CHL is a derivative of chlorophyll, which has been proven to possess detoxifying, antioxidant, chemopreventive, and antimutagenic properties. CHL is a semisynthetic mixture of copper and sodium salts derived from chlorophyll [21]. Unlike natural chlorophyll, CHL is soluble in water. This compound is used as a colorant in foods, drugs, food supplements, and cosmetics. Specifically, E141ii corresponds to copper complexes of chlorophyllins, an additive widely used in soft drinks, liqueurs, chewing gum, candies, or canned vegetables, authorized as a food additive in the European Union in accordance with Annex II of Regulation (EC) No. 1333/2008 [22]. Despite its improved stability compared to chlorophyll, the use of CHL as a colorant remains limited, since this form, like other chlorophylls and natural pigments, is susceptible to different degradation processes. The most common transformation suffered by CHL is pheophytinization, i.e., replacement of the central metal ion with hydrogen, which occurs easily under acidic conditions and exposure to mild heat [23]. In the presence of certain enzymes and high water activity, maintaining the green colors of chlorophyllin is a challenge [24]. Furthermore, although it is generally soluble in water, at acidic pH, the CHL becomes increasingly susceptible to hydrophobic aggregation as the carboxylic acids in its structure protonate, making it difficult to use in acidic beverages [25].

In this context, and in accordance with current trends in the processing of biomaterials for encapsulation of bioactives, the main objective of this work was to explore the electrospraying technique for the development of nanocapsules based on agar, with potential use in the food industry. As a strategy to avoid agar gelation and attain sufficient chain entanglements for capsule formation, acetic acid was explored as solution solvent. Furthermore, the obtained nanocapsules were evaluated to encapsulate chlorophyllin, as a case study.

## 2. Materials and Methods

### 2.1. Materials

Commercial agar was kindly provided by Hispanagar (Burgos, Spain). The agar content in the commercial agar was 80%. Acetic acid (HAc, p.a., 96% *v*/*v*), citric acid anhydrous (99% purity) and ethanol (96% *v*/*v*) were supplied by Panreac Applichem and used as received. Chlorophyllin sodium copper salt (CHL) (commercial grade) was purchased from Sigma Aldrich (Madrid, Spain).

### 2.2. Preparation of Agar Solutions

On the basis of screening studies, agar solutions of a wide range of agar solutions of different concentrations, i.e., 2.5%, 5%, 10%, 15%, and 20% (*w*/*v*), were prepared by dispersing the biopolymer in acetic acid solutions of various concentrations (from 20% to 40% (*v*/*v*)) (see Table 1). The minimum and maximum agar concentrations were fixed considering the solubility of the agar in the acid media, which provided a viscosity allowing processing. At these concentrations, agar chains can be partially hydrolyzed [20]. The initial dispersions were maintained, under stirring, at 95 °C for 2 h to induce agar solubilization, and then they were cooled down to room temperature before processing.

When CHL was incorporated for its encapsulation, it was added to the agar solutions at room temperature under magnetic stirring. Initially, 0.033% (*w*/*v*) of CHL was used for the pigment stability assays, carried out in 10% and 20% (*w*/*v*) agar solutions dissolved in 20% and 40% (*v*/*v*) HAc. Then, CHL was incorporated at a concentration of 5 wt.% of the agar content for its encapsulation by the electrospraying technique. The flow rate and the applied voltage for CHL-loaded agar solutions were 100 µL/h and 16.5 kV, respectively, as selected from preliminary tests, in order to attain stable electrospraying.

### 2.3. Characterization of the Solutions

The surface tension of the solutions was measured using the Wilhelmy method in an EasyDyne K20 tensiometer (Krüss GmbH, Hamburg, Germany) at room temperature.

The electrical conductivity of the solutions was measured using an XS Con6 conductivity meter (Labbox, Barcelona, Spain) at room temperature.

The pH of the solutions was measured using a pH meter (XS instruments).

All measurements were made at least in triplicate.

### 2.4. Electrohydrodynamic Processing of the Solutions

The agar solutions were processed using a homemade electrospinning/electrospraying equipment with a variable high-voltage (0–30 kV) power supply (Spinner-3 × -Advance, ANSTCO, Tehran, Iran). Solutions were introduced in a 5 mL glass syringes and were pumped at a flow rate of 0.15 mL/h through a stainless-steel needle (internal diameter 0.9 mm). The needle was connected through a polytetrafluoroethylene (PTFE) wire to the syringe, which was placed on a digitally controlled syringe pump, and the voltage was maintained at 15 kV. Processed samples were collected on a grounded stainless-steel plate placed at 10 cm from the tip of the needle in a horizontal configuration. The structures obtained were collected and stored under a hood to evaporate any residual solvent. All experiments were carried out at room temperature.

### 2.5. Morphological Characterization of the Particles

Scanning electron microscopy (SEM) was conducted on a Hitachi microscope (Hitachi S-4800) at an accelerated voltage of 10 kV and a working distance of 8–10 mm. The samples were previously sputter-coated with a gold–palladium mixture under vacuum (Q 150R-ES; Quorum Technologies, East Sussex, London, UK) prior to examination. Image analysis software (ImageJ) was used to determine the diameter of the particles from the SEM micrographs in their original magnification. Size distributions were obtained from a minimum of 200 measurements.

### 2.6. Stability of Agar Solutions Containing CHL

Solutions/suspensions of 0.033% *w*/*v* CHL and CHL-loaded agar solutions were prepared by dissolving 10% or 20% (*w*/*v*) of agar in 20% or 40% (*v*/*v*) HAc, respectively. HAc solutions at 20% and 40% (*v*/*v*) were also used as controls to evaluate the effect of the agar on the stability of the pigment in acidic media, where it was incorporated according to the methodology proposed by Selig et al. [25]. The stability of the solutions, stored at 40 °C, was evaluated at different time intervals, through photographic monitoring.

### 2.7. Encapsulation Efficiency

The encapsulation efficiency (EE) of the CHL-loaded agar particles was estimated by dissolving ca. 25 mg of CHL-loaded nanocapsules in 5 mL of ethanol 50% *v*/*v* by heating at 50 °C. The CHL content was analyzed by measuring the absorbance at 405 nm using UV/Vis spectroscopy (Nanodrop ND-1000, Thermo Fisher Scientific, Waltham, MA, USA). Each sample was prepared in duplicate, and triplicates were measured from each independent run, in order to obtain the mean CHL loading efficiency. The CHL loading efficiency was calculated using Equation (1).
(1)$$EE\;\left(\%\right)=\frac{Calculated\;CFL\;concentration}{Theoretical\;CFL\;concentration}\;100.$$

### 2.8. Chlorophyllin Release from the Agar Particles

For this, ca. 50 mg of CHL-loaded agar nanocapsules were suspended, in triplicate, in 10 mL of release medium and kept at 20 °C in the absence of light. Two different release media were used: 10% and 50% *v*/*v* ethanol solution, which cover the testing for foods with hydrophilic and lipophilic properties according to the Commission Regulation 10/2011 EU (10/2011/EC) [26], respectively. At different time intervals, the suspensions were centrifuged at 3500 rpm at room temperature during 10 min using a centrifuge from Labortechnik model Hermle Z 400 L (Wasserburg, Germany) and a 2 µL aliquot of the supernatant removed from sample analysis. The extracted aliquots were analyzed by UV/Vis spectroscopy (Nanodrop ND-1000, Thermo Fisher Scientific), by measuring the absorbance at 405 nm. Calibration curves for CHL quantification in the release media were previously obtained (*R^2^* = 0.99, see Appendix A). The CHL release values were obtained from two independent experiments at the same conditions. Triplicates were measured form each independent run.

### 2.9. Molecular Weight Distribution

The molecular weight of the agar was estimated by high-performance size exclusion chromatography using a Waters ACQ equipped with a Waters 2998 PAD module, a Waters 2414 refractive index detector, and a 2475 FLR module (Waters, USA). Acetic acid (1.2% *v*/*v*) was used as the mobile phase. The samples (3 mg/mL) were dissolved in the mobile phase under magnetic stirring at 40 °C, filtered through 0.22 μm pore syringe filters. Then, the samples were injected into columns in series (PolySep-GFC-P 4000 and 2000; 300 mm × 7.8 mm; Phenomenex Inc., California, USA) equilibrated at 40 °C. The injection volume was 20 μL, and the flow rate was 0.5 mL/min. Calibration was performed using pullulan standards (PSS polymer standards service GmbH, Mainz, Germany).

### 2.10. Statistical Analysis

The data obtained are expressed as the mean ± standard deviation. The statistical significance between the different samples was evaluated with an analysis of variance (ANOVA) with a Tukey test to determine the significance of the differences between the mean values (*p* < 0.05), using the free software R Project [27].

## 3. Results and Discussion

### 3.1. Optimization of the Electrospraying Process for Obtaining Agar Particles

The most important challenge when trying to obtain encapsulation structures from agar through electrohydrodynamic techniques is related to agar gelation in aqueous solutions at low concentrations. Therefore, finding suitable solvent combinations able to dissolve agar at higher concentrations and that do not gel at ambient temperature is a pre-requisite for proper processing. Agar solutions with a wide range of concentrations (from 2.5% to 20% *w*/*v*) were prepared in diluted acetic acid (20% *v*/*v*) (see Table 1). This solvent was selected aiming to avoid gelation, as a partial degradation of the agar chains has been reported to occur under acidic conditions [20]. Furthermore, acetic acid is considered suitable for food applications since it does not leave toxic residues on the dry samples [28]. The appearance of the solutions produced was different depending on the agar concentration (see Appendix A). The properties of the prepared solutions are shown in Table 1. In general, the most diluted agar solutions did not form gels in diluted acetic acid, and gelation was only observed for solutions with 15% (*w*/*v*) agar concentration or higher. Those solutions that formed gels at room temperature were, thus, not suitable for electrospraying. However, from the solutions that did not gel at room temperature, only those prepared at 10% (*w*/*v*) agar could be electrosprayed (see Appendix A). The remainder formed an unstable polymer jet that did not allow capsule formation through electrospraying. This could be ascribed to the fact that an increase in the agar concentration could induce the overlapping of molecular hydrodynamic radii. When reaching a certain concentration, sufficient polymer chain entanglements are obtained which stabilize the polymer jet allowing a good electrosprayability of the solution [29]. However, if the concentration and viscosity of the polymer solution are too high, the motion induced by the electric field will be impeded [10].

An aqueous solution containing citric acid was also evaluated as an alternative to HAc for agar solubilization. Even though a diluted citric acid (20% *w*/*v*) solution was effective to avoid gelation of 20% (*w*/*v*) agar at ambient conditions compared to diluted acetic acid (see Appendix A), it did not improve the electrosprayability of the agar solution. This could be attributed to either an extensive hydrolysis of the carbohydrate due to the acidic conditions of the citric acid solution [20], since the pH of citric acid solutions was around 1.63, or to the high conductivity of the solution (>4 mS), which impeded the stabilization of the polymer jet during electrospraying.

Considering that only the 10% (*w*/*v*) agar solution prepared with diluted acetic acid at 20% (*v*/*v*) resulted in a stable electrospraying process, two different strategies were also evaluated to understand how the solution properties affected the morphology of the structures. The first approach consisted of fixing the agar:HAc ratio at 1:2, while the second one fixed the amount of acetic acid at 40% (*v*/*v*), the minimum needed to avoid the gelation of agar at the highest concentration used. Therefore, both approaches were used for the preparation of the electrospraying solutions, and the influence of the agar and acetic acid concentration on the electrosprability and the morphology of the resulting capsules was evaluated.

The size and morphology of the electrosprayed structures are strongly dependent on the properties of the biopolymer solutions [30,31,32]. Therefore, the selected phycocolloid solutions were characterized in terms of pH, surface tension, and electrical conductivity prior to their processing, and the results are summarized in Table 2. Furthermore, Figure 1 shows the morphology of the processed structures obtained from the different phycocolloid solutions together with the particle size distributions of the dried samples.

The results revealed that solutions prepared using the first approach, increasing both the agar and the acetic acid concentration (in which the acetic acid was always twice that of the biopolymer concentration), were more adequate for the formation of agar nanostructures, showing smaller particle sizes for lower agar concentrations (10% *w*/*v*) which ranged between 50 and 400 nm. A higher particle polydispersity was observed in those prepared at 20% (*w*/*v*) agar concentration, in which the diameters ranged between 100 nm and 1 µm. Ibili and Dasdemir [33] also reported an increase in the size of nanoparticles with increasing polymer concentration. Considering the solution properties, in general, a slight increase in the conductivity of the solutions was observed with the agar concentration (cf. Table 2), whereas no significant variation was observed for the surface tension (*p* > 0.05). Therefore, differences in the particle size and sphericity of the processed solutions could be mainly attributed to changes in the rheology. Given the low pH of the solutions, the rheological properties were not evaluated to avoid instrument corrosion, but the Mw of agar in the different solutions was measured as shown in Appendix A). As expected, a certain hydrolysis of agar took place in the acetic acid solutions, and clearly defined Mw agar populations were distinguished, whose abundance depended on both the agar concentration and the acetic acid concentration. Increasing the acetic acid concentration and decreasing the agar concentration led to an increased population of short-chain agars. From the electrospraying results, it seems that there is a critical chain length required leading to biopolymer entanglement and resulting in capsule formation.

The morphology of nanostructures was detrimentally affected as the acetic acid concentration increased (40% *v*/*v*), showing signs of dripping, wetted particles, and agglomeration in those prepared with 15% (*w*/*v*) agar, probably due to a greater degradation of the polymer chains in the presence of a higher proportion of acid, which is directly related to the lower conductivity values obtained in this case and to the Mw profile shown in Appendix A). For a given acetic acid concentration, a decrease in the agar content may favor the hydrolysis of the polymer chains due to the higher acidic conditions (lower pH, see Table 1), resulting in lower molecular weight (as observed in Appendix A) and, thus, decreasing the formation of chain entanglements, which is known to be the main factor determining nanostructure formation. This would explain the fact that solutions prepared with 10% (*w*/*v*) agar at the highest acetic acid concentration could not be electrosprayed.

Therefore, one of the solution parameters that is crucial for the electrospraying processing of agar solutions is the concentration of the acetic acid solution used as a solvent. It seems that the acetic acid concentration should be twice more than the agar concentration for the optimal electrospraying, showing the most spherical morphology; thus, these were selected to load a bioactive compound. As the surface tension of the agar solutions did not significantly vary with the acetic acid concentration (*p* > 0.05), the dripping in those prepared with higher proportion of acetic acid could be attributed to a slight decrease in the viscosity of the solutions (as deduced from the molecular weight of the agar, see Appendix A) in combination with the decrease in their electrical conductivity, and ascribed to the higher degree of biopolymer chain hydrolysis.

### 3.2. Stability of Chlorophyllin in Agar Solutions

Once the conditions for the production of electrosprayed agar capsules were optimized, solutions obtained with the lowest and highest agar concentration were firstly selected to load a model bioactive compound and to test the stability of dispersed CHL in the agar solutions. To this end, an accelerated stability test was conducted at mild heating conditions (40 °C) with the agar solutions being stored for 7 days, and the results are shown in Figure 2.

The first thing to highlight is that the CHL rapidly began to aggregate and settled down to the bottom of the control containers (containing the solvent solutions without agar) prepared with 20% and 40% (*v*/*v*) acetic acid. In contrast, the addition of agar counteracted the aggregation phenomenon, showing an improvement in the long-term stabilization of the pigment’s solubility in acid solutions even after 7 days at 40 °C. It is possible that the viscosity provided by the agar continuous phase improved the stability against CHL aggregation, significantly reducing the aggregation of the bioactive. This improvement was greater in those prepared with a higher agar concentration.

Selig et al. [25] explored the hypothesis that long-chain anionic polysaccharides, particularly xanthan gum and sodium alginate, can interact through the negative charges on the polymer chain with the positive CHL. These small electrostatic interactions may be enough to stabilize the solubility of pigments in acidic solutions and improve color stability by reducing water activity, which affects access to enzymatic degradation reactions and improves copper ion stability, thus improving the color in the core of the pigment structure. These authors evaluated the effects of xanthan gum and alginate in aqueous solutions of CHL with citric acid. On the basis of this work, it would be interesting to evaluate the behavior of CHL in agar nanocapsules, thus guaranteeing the stability of the pigment in acid solutions. Furthermore, due to their low stability at high temperatures, the use of the electrospraying technique is interesting for these types of heat-sensitive additives.

### 3.3. CHL Release from the Electrosprayed Agar Particles

Once the conditions for the production of electrosprayed agar particles were optimized and the stability of the bioactive in the agar solutions was proven, agar suspensions with the higher biopolymer concentration were loaded with CHL since the stability of the bioactive was increased at higher agar concentration, and increasing the amount of biopolymer in the electrospraying solution would increase the yield of the resulting electrosprayed capsules. To this end, the agar solution (20% *w*/*v*) was dissolved in 40% (*v*/*v*) acetic acid solution, as in previous experiments, and CHL was subsequently incorporated at room temperature to achieve a final theoretical concentration of 2.5 or 5 wt.% of the agar content. In both cases, the EE was relatively low, showing no significant differences: 38.2 ± 0.8 and 38.6 ± 0.7 for 2.5 and 5 wt.%, respectively. This effect could be either ascribed to a partial degradation of nonencapsulated CHL or to a partial precipitation of the CHL in the syringe during the electrospraying process. Furthermore, the composition and type of wall and core material influence the encapsulation efficiency [34]. Higher encapsulation efficiencies (>80%) have been reported for hydrophilic active compounds encapsulated into hydrophilic matrices such as chitosan or gelatin [35,36]. In contrast, lower encapsulation efficiencies (between 23–60%) were found when hydrophilic matrices (i.e., proteins and polysaccharides) were loaded with hydrophobic compounds [37]. Figure 3 shows a representative micrograph of the agar microcapsules loaded with 5 wt.% CHL, which exhibits a similar morphology to that obtained in the absence of the CHL.

The release profile of encapsulated bioactive compounds is of outmost importance when designing delivery systems. Therefore, a release study of CHL from the electrosprayed agar particles was carried out in two different food simulants with different hydrophilicity (10% *v*/*v* and 50% *v*/*v* ethanol), according to the Commission Regulation 10/2011 EU (10/2011/EC) [26], which simulate hydrophilic and lipophilic foods, respectively. The resulting release profile of CHL soluble in 50% (*v*/*v*) ethanol (the medium in which the release was maximum) is depicted in Figure 4.

The release of CHL from the capsules in the medium with lower ethanol concentration was negligible, which could be ascribed to a lower solubility of the bioactive in this medium. Another possibility can be related to the effect that the presence of ethanol has on the water network of a polysaccharide with gelling properties [38,39], which directly affects the biopolymer network interactions and, thus, the stability of gel structure. Briefly, the presence of a higher concentration of alcohol tends to destabilize the hydrogen bonds of gels (as also reported for gellan gum or κ-carrageenan by Cassanelli et al. [38]), thus decreasing the network order. In contrast, the presence of higher concentrations of ethanol could improve the solubility of CHL; however, as commented above, it could also destabilize the hydrogen bonds, affecting the mobility of the biopolymer chains and promoting less hydrated agar capsules with higher porosity and a faster CHL release. As a result, an initial burst release was observed, followed by a slower sustained release when the capsules were immersed in 50% ethanol solution.

## 4. Conclusions

Food-grade agar-based encapsulation matrices were, for the first time, successfully developed by electrospraying agar solutions in diluted acetic acid. The electrospraying process was initially optimized in order to obtained neat capsules, without fibrils. The morphology of the nanostructures was detrimentally affected as the acetic acid concentration increased (40% *v*/*v*), showing signs of dripping, wetted particles, and agglomeration in those prepared with 15% (*w*/*v*) agar, due to a greater degradation of the polymer chains in the presence of a higher acid concentration. The results also showed that solutions prepared by increasing both the agar and the acetic acid concentration (in which the acetic acid doubled the biopolymer concentration) were more adequate for the formation of agar nanostructures, showing smaller particle sizes for lower agar concentrations (10% *w*/*v*) which ranged between 50 and 400 nm.

Electrosprayed agar particles with theoretical CHL loading of 5% *w/w* were developed, and the release of CHL from the agar nanocapsules was also tested. With an estimated encapsulation efficiency of around 40%, CHL release was significantly higher in the 50% ethanol solution, being almost negligible in the 10% ethanol one. Future work to reach higher encapsulation efficiencies should be conducted to further confirm their suitability as delivery vehicles.

## Figures and Tables

**Figure 1 foods-11-02093-f001:**
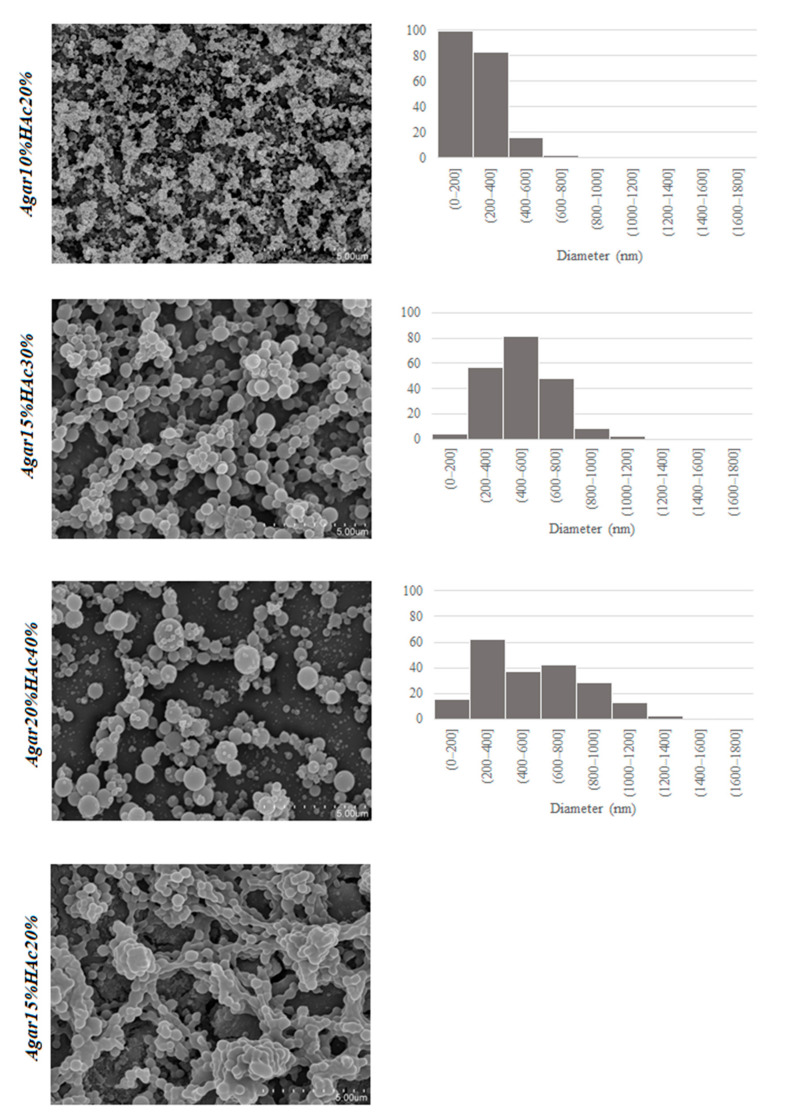
SEM images of the agar structures obtained through electrospraying from different agar and acetic acid concentration (**left**) and particle size distributions for the electrosprayed samples (**right**) obtained from measurements of 200 capsules. Scale bars in SEM images correspond to 5 µm.

**Figure 2 foods-11-02093-f002:**
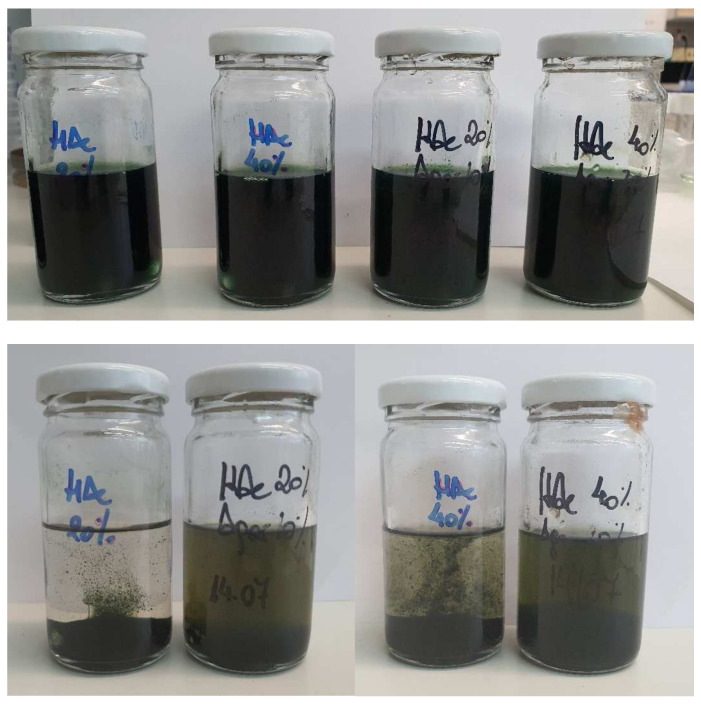
CHL stability test in fresh acetic acid solutions (**up**) and stored at 40 °C, for 1 week (**down**). From left to right: acetic acid 20% *v*/*v* (*HAc20%*)*,* agar 10% *w*/*v* in acetic acid 20% *v*/*v* (*Agar10%HAc20%*), acetic acid 40% *v*/*v* (*HAc40%*), and agar 20% *w*/*v* in acetic acid 40% *v*/*v* (*Agar20%HAc40%*).

**Figure 3 foods-11-02093-f003:**
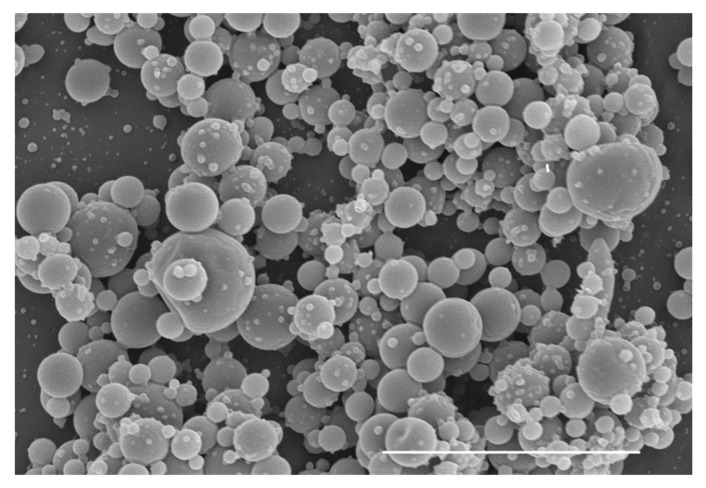
Electrosprayed agar nanocapsules loaded with 5 wt.% CHL of the agar content (scale bar corresponds to 4 μm).

**Figure 4 foods-11-02093-f004:**
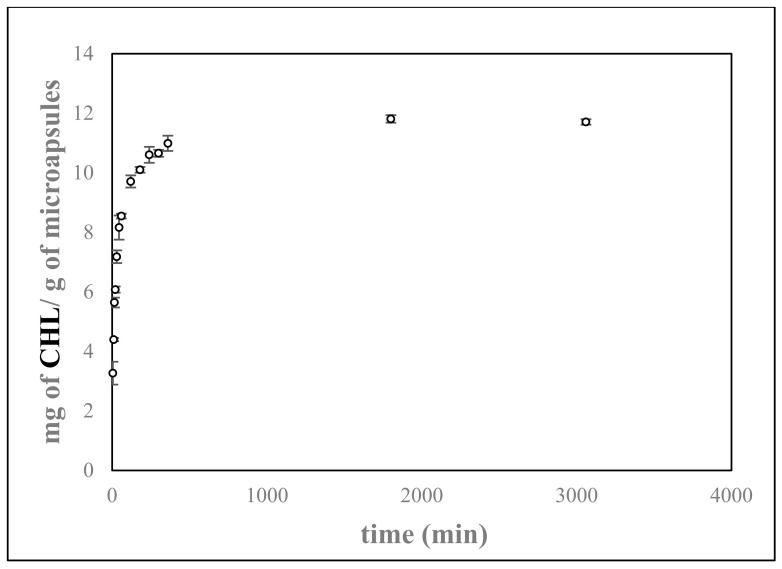
Chlorophyllin (CHL) release from electrosprayed agar nanocapsules in 50% ethanol.

**Table 1 foods-11-02093-t001:** Gelling and electrosprayability of different agar solutions.

Sample	Agar (% *w*/*v*)	Acetic Acid (% *v*/*v*)	Citric Acid (% *w*/*v*)	Gelling at Room Temperature	Electrosprayability
**1**	2.5	20	0	non-gelling	−
**2**	5	20	0	non-gelling	−
**3**	10	20	0	non-gelling	+
**4**	15	20	0	gel	− −
**5**	20	20	0	gel	− −
**6**	20	0	20	non-gelling	−

+: *electrosprayable biopolymer solution*. −: *unstable biopolymer jet*.
− −: *not tested for electrosprayability (gelled at room temperature)*.

**Table 2 foods-11-02093-t002:** Properties of the different acetic acid agar solutions.

*Sample*	pH	Electrical Conductivity (µS)	Surface Tension (mN/m)	Electrosprayability
** *Agar10%HAc20%* **	2.45 ± 0.02 ^a^	1469 ± 9 ^ab^	39.2 ± 0.6 ^a^	+
** *Agar15%HAc30%* **	2.37 ± 0.03 ^b^	1481 ± 34 ^a^	39.0 ± 1.1 ^a^	+
** *Agar20%HAc40%* **	2.31 ± 0.03 ^c^	1521 ± 147 ^a^	39.2 ± 0.8 ^a^	+
** *Agar15%HAc40%* **	2.22 ± 0.01 ^d^	1335 ± 44 ^b^	37.1 ± 1.3 ^a^	+
** *Agar10%HAc40%* **	2.07 ± 0.02 ^e^	1118 ± 16 ^c^	36.9 ± 2.7 ^a^	−

Mean values ± standard deviation. Different letters in the same column indicate significant differences between samples (*p* < 0.05). +: *electrosprayable biopolymer solution*. −: *unstable biopolymer jet*.

## Data Availability

Data are available in a publicly accessible repository. The data presented in this study are openly available from DIGITAL.CSIC at https://doi.org/10.20350/digitalCSIC/14599 [40].

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
