# Peer review of "Electrosprayed Agar Nanocapsules as Edible Carriers of Bioactive Compounds"

_foods, 2022, doi:10.3390/foods11142093_

Round 1

Reviewer 1 Report

The article entitled "ELECTROSPRAYED AGAR NANOCAPSULES AS EDIBLE CARRIERS OF BIOACTIVE COMPOUNDS, aimed to produce agar nanocapsules using an acetic acid solution as a solvent. The report was well written and brings a great new material to produce nano composites using electrospray methodology. The introduction provides relevant background and includes updated references contextualizing the study in the research field. Material and methods bring all the methods utilized, and they are well described and show the appropriate techniques to approach this kind of work. The statistical analysis was also appropriated.

The results are well discussed, the figures and tables are appropriate, and the conclusion is in accordance with the objectives.

Reviewer 2 Report

Overall

The topic is of interest to the scientific community and is timely. However, the introduction part must be improved by further elaborating the various merits and demerits of using electrospinning and electrospraying techniques and why author has chosen electrospraying technique for his study. In addition, add previous related reported work and then explain the gap and what your plans to fill it. In-addition author has not explained why they have chosen acetic acid as a solvent must explained in introduction part. The technical quality of the manuscript in present form is not suitable for publication and must be improved and add more data to support findings.  

Introduction

The topic is of interest to the scientific community and is timely. However, the introduction part must be improved by further elaborating the various merits and demerits of using electrospinning and electrospraying techniques and why author has chosen electrospraying technique for his study. As this part must be improved to explain the background. What is already done and why is this study being carried out? Otherwise, the novelty of this research will be question mark. In addition, add previous related reported work and then explain the gap and what your plans to fill it. Moreover, it would be better and recommended that abbreviation should be explained first and then used. Such as PTFE in line 133, ca in 156 check this through the manuscript, and why in some place’s ca is italic and in some plcace it is not. (156 and 163 lines).

Materials and Methods

Provide further details of all the material used in the manuscript along with purity and product codes of all these materials and the chemicals. As MDPI has policy that all the reported results must be reproducible and can be verify. This is serious drawback of this research work. So You must be precise when writing protocols; everyone should be able to repeat your experiments, after this paper is published, and gain the same results. So, please add appropriate references which are missing in Materials and Methods section. Who exactly is the author of the methods applied? So Appropriate references to the methods should be provided. And the statistical findings have to be given in the text such as (p<0.05) or (p>0.05).

Results and discussion

In my opinion, the author should explain their results and findings in more detail and recent references should be added. The overall discussion part lacks the reasoning and comparison with previous reported work. Are the data presented in figures significantly different? At least error bars should be shown i.e., figure 1 (right). All figures must be self-explanatory. And the results and discussion are not enough to prove the conclusion and I would suggest rewrite the conclusion with clear novelty.

Reviewer 3 Report

 Abstract

-          it is interesting to add values and comparisons in the abstract

-          -how about the major results of this study

Introduction

-          L34-35, this sentence should be developed with appropriate and recent reverences

-          L37-38, “and can interact with other compounds in the food matrix leading to a loss in the quality of functional assets” what are type of interactions, and how the quality will reduce?

-          L50-51, how about the applicability of described process in food industry

-          Please add some examples of bioactive compounds encapsulated by electrohydrodynamic processing

-          L61-65 long sentence,

-          great mechanical strength with moderate water resistance and good barrier properties: this sentence was important to develop it specially describe the importance of agar traits to be encapsulated by  electrohydrodynamic

-          Please re write the aim of your study

MATERIALS AND METHODs section

-          L104, what is the purity of agar used in this study

-          L105 please indicate the choice of the used concentrations

-          Section 2.2 need a reference

RESULTS AND DISCUSSION section

-          L198 (see Table 1-> Table 1

-          L 200-201 what is the proper concentration of acetic acid in food industry without toxic impact

-          Table 1 was not clear, what is meaning of Electrosprayability + and -, more explain this fact in the MS

-          Table 2, the superscripts are not appropriate in the pH column (e.g)

-          Fig 1, superscripts are missed7

-          At 40% of EE%, I think it small compared with other works, please explain this fact

-          Delete the fig 2, poor quality

-          Authors should compare their work with other’s with recent references

-          All data in this MS could be correlated when authors used several trials

-         

Round 2

Reviewer 2 Report

The Author has improved the manuscript and can be accepted in the present form.

Reviewer 3 Report

this MS can be accepted and published